No pump, no problem: evaluating passive eDNA sampling for marine biomonitoring of a nuisance macroalga

Nichols Patrick K. patrick.kendall.nichols@gmail.com
Marko Peter B.
School of Life Sciences, University of Hawaiʻi at Mānoa , Honolulu , HI , United States of America
Costantini Federica
Electronic publication date: 2025 Aug 25
Publication date: 2025
Volume: 13
Electronic Location ID: e19939
Received 2025 Mar 28; Accepted 2025 Jul 27
Copyright: ©2025 Nichols and Marko
Copyright year: 2025
Copyright holder: Nichols and Marko
License: This is an open access article distributed under the terms of the Creative Commons Attribution License, which permits unrestricted use, distribution, reproduction and adaptation in any medium and for any purpose provided that it is properly attributed. For attribution, the original author(s), title, publication source (PeerJ) and either DOI or URL of the article must be cited.
License URL: https://creativecommons.org/licenses/by/4.0/

Keywords: Molecular detection, Chondria tumulosa, Coral reefs, Occupancy modeling, Papahānaumokuākea Marine National Monument

Funding: The University of Hawaiʻi at Mānoa Charles H. and Margaret B. Edmondson Research Fellowship and Melanie F.G. Brose Eco-Policy Fellowship Funding was provided by the University of Hawaiʻi at Mānoa Charles H. and Margaret B. Edmondson Research Fellowship and Melanie F.G. Brose Eco-Policy Fellowship. The funders had no role in study design, data collection and analysis, decision to publish, or preparation of the manuscript.

==============================
Efficient detection and management of non-indigenous species are critical for mitigating their ecological impacts. Environmental DNA (eDNA) techniques have transformed biodiversity monitoring by enabling sensitive and cost-effective surveys. This study compares the efficacy of passive eDNA samplers (PEDS) to conventional active filtration methods for detecting the cryptogenic macroalga Chondria tumulosa within the Papahānaumokuākea Marine National Monument, Hawaiʻi, USA. Three components of the species-specific eDNA assay were evaluated: clinical sensitivity, DNA yield, and overall performance. Site-level detection sensitivity of 15-minute PEDS deployments matched that of 2-L active filtration, with both methods detecting C. tumulosa in all cases where it was known to present. Site-occupancy models provided a robust framework for evaluating overall performance, offering critical insights into the tradeoffs of PEDS for detecting rare taxa. The success of PEDS is largely dependent on the increased number of qPCR replicates employed in this study compared to the previously developed eDNA assay for C. tumulosa. Passive method performance resulted in lower qPCR detection rates with higher probabilities of false-positives and false-negatives. Model estimates for C. tumulosa eDNA occupancy were similar between PEDS affixed to stationary buoys and PEDS attached to roving SCUBA divers. There was, however, a decrease in the eDNA capture rate among samples collected while on SCUBA. We also tested two passive membrane types—research-grade mixed cellulose ester filters and low-cost cotton rounds. The absorbent cotton rounds yielded greater target eDNA yields and were more reliable for inferring the presence of C. tumulosa. However, DNA yields from PEDS were consistently lower than actively filtered samples, indicating the importance of optimizing sampling and processing protocols to balance erroneous detections. Despite these limitations, passive sampling successfully detected C. tumulosa at low abundances (<1%), demonstrating its utility for uncovering cryptic taxa. PEDS are a cost-effective, versatile, and scalable alternative to active filtration, particularly in remote or resource-limited settings.

Introduction

Non-indigenous species (NIS) pose a substantial threat towards ecologically sensitive and culturally important habitats (Olden et al., 2004; Molnar et al., 2008; Ehrenfeld, 2010). Oftentimes management decisions require a rapid understanding of a species’ distribution to implement effective eradication or containment strategies (Bott et al., 2010). However, for taxa that can disperse widely, monitoring vast areas is often prohibitive, particularly when relying on visual detection methods (Pereira & Cooper, 2006; Campbell, Gould & Hewitt, 2007; Yoklavich, Reynolds & Rosen, 2015). Bio-surveillance of rare taxa can be enhanced with molecular approaches, such as the analysis of environmental DNA (eDNA), any genetic material shed by an organism into its environment (e.g., water, soil, air) (Jerde et al., 2011; Goldberg et al., 2013; Taberlet et al., 2018). eDNA approaches can be more sensitive, efficient, and cost-effective than standard visual-based survey methods (Kelly et al., 2014; Smart et al., 2015; Smart et al., 2016; Blackman et al., 2020; Keller et al., 2022). As a result, eDNA tools have been gaining traction as an effective method for monitoring NIS (Darling & Blum, 2007; Borrell et al., 2017; von Ammon et al., 2023).

Active filtration is a widely used method for collecting eDNA from aqueous environments by forcing known volumes of water through filter membranes. This technique concentrates eDNA within a porous membrane, allowing for detection of rare taxa and cryptic life forms (Harvey, Hoy & Rodriguez, 2009; Doyle, McKinnon & Uthicke, 2017; Keller et al., 2022; Nordstrom et al., 2024). By capturing genetic material from large volumes of water, active filtration enhances the sensitivity of molecular detection methods (Liang & Keeley, 2013; Tsuji et al., 2019; Shu, Ludwig & Peng, 2020). Additionally, innovative applications have expanded its utility, including fully-contained manned filtration units (Thomas et al., 2018) and remotely deployed robotic pumps, which enable efficient sample collection in various environments (Formel et al., 2021; Hendricks et al., 2023). When on-site processing is not possible, water samples can also be frozen or refrigerated before filtration, providing some flexibility in field applications (Hinlo et al., 2017; Sales et al., 2019).

Despite its advantages, active filtration comes with several challenges. Filtering large volumes of water can be labor-intensive and expensive, potentially limiting the sampling capacity of eDNA studies, particularly in remote locations (Jeunen et al., 2022; Yan et al., 2024). These methods, while reliable, depend on specialized equipment and dedicated laboratory space (Murray, Coghlan & Bunce, 2015; Goldberg et al., 2016; Tsuji et al., 2019). Active filtration of water necessitates technical expertise and a strict adherence to laboratory best-practices as the process risks contamination from handling collection containers, filtration gear, and laboratory equipment—a critical consideration in any eDNA study (Murray, Coghlan & Bunce, 2015; Goldberg et al., 2016; Djurhuus et al., 2017; Larson et al., 2020). Perhaps the most limiting step for conducting widespread eDNA sampling is the filtration of water, which can take 10–40 min or more per replicate, depending on the volume sampled, filter pore size, and water turbidity (Toshiaki et al., 2020; Zaiko et al., 2022; Wu et al., 2024; Yan et al., 2024). Actively pumping water through membranes can be time-prohibitive for rapid biomonitoring, particularly during short-term sampling efforts, such as during remote expeditions. eDNA methods for routine biomonitoring benefit from maximizing replicate sample collection for ecological inferences (MacKenzie, 2005; Bessey et al., 2021) while minimizing unnecessary costs, equipment, and personnel (Dickie et al., 2018).

To address active filtration challenges, passive eDNA collection, where membranes are simply exposed to eDNA in situ, has been introduced as a simpler biomonitoring approach (Bessey et al., 2021; Chen et al., 2024). Passive environmental DNA samplers (PEDS) eliminate the need for water filtration and can therefore both increase sampling capacity and minimize contamination compared to conventional active filtration (Bessey et al., 2021). In aqueous environments, passive collection relies on eDNA adhering to the filter membrane as water flows around it over time. Exposure duration, porosity, membrane surface area, and physio-chemical properties of the water influence the passive recovery of eDNA (Liang & Keeley, 2013; Kirtane, Atkinson & Sassoubre, 2020; Bessey et al., 2021; Bessey et al., 2022; Maiello et al., 2022). In a study of aquarium fish in a large mesocosm, absorbent materials were able to detect the majority of taxa in short 5 min exposures (Bessey et al., 2022). However, DNA yield is often lower with PEDS (Kirtane, Atkinson & Sassoubre, 2020; Bessey et al., 2021; Van der Heyde et al., 2023), and retrieving membranes risks eDNA loss, as adsorbed material is not actively secured to the membrane (Jeunen et al., 2022). Therefore, the utility of PEDS for reliably detecting rare taxa remains limited (Nordstrom et al., 2024).

Rare and cryptic taxa are especially prone to detection errors due to low abundance, stochastic environmental factors, and methodological limitations (MacKenzie, 2005; Pearman et al., 2016). Consequently, many NIS may go undetected as “sleeper populations” (Spear et al., 2021). Molecular detections of rare taxa depend on the platform used (e.g., high-throughput metabarcoding or quantitative polymerase chain reaction, qPCR), habitat being sampled, and organisms being targeted (Bessey et al., 2021; Jeunen et al., 2022; Verdier et al., 2022; Zhang et al., 2024). For instance, metabarcoding of eDNA with universal fish primers showed comparable or greater richness estimates between passive and active eDNA collection at temperate coastal sites (Bessey et al., 2021; Zhang et al., 2024). However, at tropical sites passive and active collection methods recovered distinct communities with little overlap (Bessey et al., 2021). Species-specific qPCR assays have been implemented for NIS detection across terrestrial, freshwater, and marine systems (Doyle, McKinnon & Uthicke, 2017; Leblanc et al., 2020; Valentin et al., 2020; Keller et al., 2022; Bommerlund et al., 2023; Villacorta-Rath et al., 2023) because they are particularly sensitive to trace amounts of target DNA present in a sample (Rasmussen, 2001; Bohmann et al., 2014; Nathan et al., 2014; Taberlet et al., 2018). The sensitivity of this method can signal NIS presence when visual detections would otherwise be improbable (Kim & Byrne, 2006; Darling & Blum, 2007; Harvey, Hoy & Rodriguez, 2009) and may be better suited for detection of rare taxa using PEDS.

Remote island reserves are particularly vulnerable to NIS (Vitousek, 1988; Moser et al., 2018), yet are difficult to routinely survey (Wilhelm et al., 2014). Papahānaumokuākea Marine National Monument (PMNM), a UNESCO world heritage site, is threatened by the cryptogenic red alga Chondria tumulosa (Rhodomelaceae, Rhodophyta), first observed smothering extensive areas of the reef at Manawai (Pearl and Hermes Atoll) in 2016 (Sherwood et al., 2020), and subsequently discovered at neighboring Kuaihelani (Midway Island) and Hōlanikū (Kure Atoll). Although visual surveys are ideal to establish the presence of C. tumulosa, the size and remoteness of PMNM’s reefs make early visual detection difficult, potentially delaying decision-making by managing stakeholders. As NIS become well-established, management costs rise substantially (Leung et al., 2002). Sensitive biomonitoring tools are critical to ensure comprehensive and feasible detection of rare taxa given logistical constraints. 

In this study, we test the applicability of PEDS using a qPCR assay specifically developed for C. tumulosa eDNA. First, we assessed clinical sensitivity (i.e., the proportion of correctly identified positive site-level detections) and DNA yields between PEDS using relatively short exposure times (15 min) and conventional active filtration of 2-L seawater samples. Second, we compared conventional mixed cellulose ester (MCE) analytical filters and a more cost-effective cotton alternative as membranes for use in PEDS. Finally, we implemented site-occupancy detection modeling to estimate assay overall performance (hereafter referred to as “performance”) and error rates among PEDS deployments on stationary buoys and roving SCUBA divers. Our findings indicate that PEDS had comparable detection dynamics for C. tumulosa and represents a promising method to increase the utility of NIS biomonitoring in remote, ecologically sensitive coral reefs, such as those protected within PMNM.

Materials & Methods

Field work was conducted under Papahānaumokuākea Marine National Monument Co-Trustees permits PMNM-2023-01 and PMNM-2024-001. Thirty-four sites from within three atolls in PMNM were sampled for C. tumulosa eDNA between 2023–2024 (Table S1). Manawai and Kuaihelani are atolls with documented high abundances of C. tumulosa (Sherwood et al., 2020; Lopes Jr et al., 2023; Fraiola et al., 2023). Hōlanikū represents an emerging colonization front where C. tumulosa is growing cryptically (Nichols et al., 2025). At each site, experts in the morphological identification of C. tumulosa estimated benthic cover within a 10 m radius (∼314 m2), centered at the location of eDNA collection. Given the patchy distribution of C. tumulosa in this region, combining visual data with small-scale variations in target species cover and environmental covariates provides the greatest potential to inform the model (see Site-occupancy detection modeling, below).

Methods comparison

Passive vs. active collection of eDNA

PEDS were constructed using polyester mesh screening (mesh opening: two mm), secured on three sides with cable ties (Fig. 1). A subset of five sites from Hōlanikū and five from Kuaihelani were used to compare passive and active filtration methods (Table S1). For active filtration, surface seawater samples were pumped through mixed cellulose ester (MCE) filter membranes (Millipore Durapore, diameter: 45 mm, pore size: 0.22 µm) on duplicate 47 mm filter holders (Advantec MFS, Inc., Irvine, CA, USA), using a portable peristaltic pump with dual pump heads (Proactive Environmental Products, Bradenton, FL, USA). Pump tubing with a 0.5 kg lead weight at one end was deployed directly into the surface water from an anchored small vessel to a depth of one m. Water was pumped through MCE filters at a rate of 0.13 L min−1 until a total of 2-L was filtered per replicate. Simultaneously, MCE filter membranes secured in PEDS were suspended in the water column directly below the pump tubing and were removed after 15 min of exposure. Depths at these sites were limited to <4 m, ensuring that active and passive collections occurred both within four m of the benthic habitat and within 0.5 m of each other.

Figure 1 Passive environmental DNA samplers (PEDS).

(A) PEDS were constructed using mesh screen pouches that were then loaded with either mixed cellulose ester (MCE) analytical filters or 100% cotton rounds. PEDS were then either (B) suspended stationary in the water column on a weighted buoy or (C–D) attached to roving surveyors. Photo credit: Patrick Nichols and Nathan Eagle/Civil Beat.

Passive membrane material

Passive eDNA detection was compared between MCE analytical filters and 100% cotton rounds (ASIN: B09542G9ZN, approximate diameter: 50 mm; Amazon, Seattle, WA, USA) at 12 sites from Kuaihelani (Table S1). Cotton rounds were selected for their low cost and comparable size to MCE filters. Duplicates of each cotton and MCE membranes were attached to weighted surface marker buoys (Fig. 1), ensuring consistent positioning above the benthic habitat in the bottom one m of the water column for 15 min.

Passive roving divers

At Manawai, where surface marker buoys were not able to be deployed, duplicate cotton rounds in PEDS were instead attached to a single SCUBA diver conducting visual assessments (Fig. 1). PEDS were removed immediately after surfacing, with membrane exposure time coinciding with the total dive time (surface-to-surface) at each of the 12 sites (Table S1).

Contamination prevention

To reduce contamination risks, all sampling equipment (mesh PEDS, float buoys, pump tubing, MCE filter holders) were decontaminated by soaking in a 10% bleach solution for at least one hour and then randomly shuffled for re-use as both biological replicates and negative control equipment and extraction blanks (EB). Furthermore, peristaltic pump tubing was continuously flushed with bleach for 5 min and dried prior to use. Gloves were worn and bleach-decontaminated forceps were employed for handling each filter membrane. For all passive sampling, a negative control EB filter membrane was brought into the field (one per day) and processed alongside biological samples, serving as a negative control for the eDNA collection and extraction protocol. For active filtering of eDNA, a 1-L tap water EB was brought along into the field and filtered on decontaminated equipment before biological samples. DNA extraction and qPCR amplification were performed in physically separate laboratory spaces where all surfaces and equipment were routinely decontaminated with a 10% bleach solution.

DNA extraction

Following eDNA collection, all membranes were stored on ice in single-use, sealable plastic bags for no longer than 6 h before being stored at −20 °C until laboratory processing. DNA was extracted using commercially available kits (DNeasy Blood & Tissue kit; Qiagen, Germantown, MD, USA) from a one cm strip (∼1.5 g wet weight) cut from the center of thawed passive membranes. Active filters were cut in half, with only one half undergoing DNA extraction. All membranes were extracted using the manufacturer’s protocol with the following modifications: no Proteinase K incubation step (Timmers et al., 2024), use of 600 µL ATL Buffer, 600 µL AL Buffer, 600 µL ethanol, and two 50 µL AE Buffer elutions, resulting in a final volume of 100 µL extracted DNA.

qPCR amplification

Quantitative PCR conditions followed those described in our previous study developing the qPCR assay to target a 95-bp rbcL gene fragment from C. tumulosa (GenBank ID: MT039604; Nichols et al., 2025), with adjustments to accommodate the low-yield eDNA often obtained from passive membranes (Kirtane, Atkinson & Sassoubre, 2020; Bessey et al., 2021; Van der Heyde et al., 2023). Each reaction contained 4.5 µL of SSoAdvanced SYBR Green Supermix (Bio-Rad), 0.5 µL of bovine serum albumin (BSA) to reduce inhibitors (Wilson, 1997), 0.1 µL of each forward (5′-GCCGTGAATCGTTCTATTGC-3′) and reverse (5′-TCAGCTCTTTCGTACATATTCTCC-3′) primers (10 µM), and 4.8 µL of extracted DNA, for a total volume of 10 µL. qPCR was performed on a CFX96 Touch system (Bio-Rad) with the following thermal cycling conditions: initial activation at 95 °C for 3 min, followed by 10 cycles of denaturation at 95 °C for 30 s, annealing at 67 °C for 30 s (decreasing by 1 °C per cycle), and extension at 72 °C for 30 s. This was followed by 30 cycles of denaturation at 95 °C for 30 s, annealing at 54.7 °C for 30 s, extension at 72 for 30 s, with a final extension at 72 °C for 10 min, and a melt curve analysis from 65–95 °C (increasing by 1 °C per cycle). To account for the potential lower detection rates with passive membranes, six technical qPCR replicates were used for each sample (Stevens et al., 2024). Critical thresholds of quantification (Nq, above which detections can be discriminated from background noise) and Cq values were automatically calculated in the CFX Manager software (v3.1) using the regression determination mode, with no outliers removed.

Each PCR plate included triplicate serial dilutions of C. tumulosa DNA standards (10−1 to 10−7 ng µL−1) to generate a regression line for determining the run efficiency (E, calculated automatically in the CFX Manager software). Efficiency-corrected eDNA starting quantities (N0) were calculated for each replicate within a 96-well plate using the formula: N0= Nq/ECq, where Nq is the fluorescence quantification threshold, E is the efficiency, and Cq is the quantification cycle, thus normalizing sample concentrations across plates with varying efficiencies (Ruiz-Villalba, Ruijter & Van den Hoff, 2021). Triplicate no-template controls were included on each qPCR plate to monitor for reagent contamination. Prior to running the assay, we screened for PCR inhibition by comparing the standard curve efficiencies of C. tumulosa synthetic DNA (106–101 copies per µL) against the same synthetic DNA spiked into a subset of sample eDNA matrix extracts (n = 7, 1.21 ×10−1−1.21 ×10−7 ng per µL). Inhibition was assumed if eDNA matrix Cq values were significantly greater than those expected based on the synthetic standard curve. Internal positive controls (IPCs) were not used to avoid introducing additional contamination risk across low-template eDNA samples, especially given the high-throughput nature of this study. This approach aligns with MIQE recommendations (Bustin et al., 2009), which emphasize the value of dilution-based methods for evaluating inhibition.

Data visualization and statistical analyses

Raw qPCR amplification and melt curve data were visualized using the ggplot2 package (Wickham, 2011), fitted with generalized additive model (GAM) smoothers in R v4.2.3 (R Core Team, 2024). Sites were assigned a naïve positive detection if the best-fit GAM smoother exceeded the quantification threshold (Nq), demonstrating precision across technical replicates. Wilcoxon signed-rank tests were used to compare efficiency-corrected starting quantities (N0) between paired methods (active/passive collection and cotton/MCE membranes).

Site-occupancy detection modeling

Site-occupancy detection models offer a statistical framework to address challenges with imperfect observations, estimating detection probabilities and associated error rates (MacKenzie et al., 2002; Royle & Link, 2006; Guillera-Arroita et al., 2017). These models can rigorously evaluate assay performance, ultimately guiding survey methodology selection for routine biomonitoring (Schmidt et al., 2013; Doi et al., 2019; Peixoto et al., 2021; Buxton et al., 2021). We applied a Bayesian hierarchical site-occupancy detection modeling framework to qPCR detection data (Schmidt et al., 2013; Lahoz-Monfort, Guillera-Arroita & Tingley, 2016; Griffin et al., 2020), using the eDNA RShiny application (Diana et al., 2021, https://seak.shinyapps.io/eDNA/) for R, which allows augmentation with opportunistic visual surveys and accounts for both false-positive and false-negative errors (Lahoz-Monfort, Guillera-Arroita & Tingley, 2016; Guillera-Arroita et al., 2017; Griffin et al., 2020). Site-level eDNA presence-absence was inferred through two key metrics: the posterior probability of presence and occupancy. The posterior probability of presence indicates the likelihood that a site is occupied based on eDNA qPCR detection data, opportunistic visual survey data, and habitat covariates. If the target organism is visually observed, the posterior probability of presence is fixed (presence = 1), but visual absences do not override eDNA-based presence estimates. The occupancy probability infers the inherent probability that a site is occupied due to its site-specific characteristics. We therefore focus on the probability of presence to inform C. tumulosa presence-absence and use the occupancy probability to draw inferences from habitat covariates included in the model.

Using only data from PEDS, posterior probabilities of C. tumulosa presence and occupancy at a site (ψ), a sample from an occupied site containing C. tumulosa eDNA (θ11), and a positive qPCR replicate from a sample containing C. tumulosa eDNA (p11) were modeled as a function of habitat covariates. Continuous covariates included site depth, C. tumulosa abundance (from visual surveys), X-Y site position coordinates, and membrane exposure time in the water column. Continuous covariates were standardized by subtracting the mean and dividing by the standard deviation for each covariate. We computed Pearson’s correlation coefficients for all pairs of continuous covariates using the cor function in R. A correlation threshold of ± 0.7 was used to identify collinearity between covariates. In cases of strong correlation, one variable from each correlated pair was excluded from the analysis. We also included PEDS deployment method as a categorical covariate with three levels: cotton rounds on a stationary buoy (“Cotton/Buoy”), MCE membrane on a stationary buoy (“MCE/Buoy”), or cotton rounds attached directly to survey divers (“Cotton/SCUBA”). False-positive inferences at the sample level (θ10), false-positive tests at the qPCR replicate level (p10), and false-negative tests (1 − θ11 and 1-p11, respectively) were also estimated. The assay’s ability to infer eDNA presence-absence from qPCR replicates was estimated using the conditional probability of C. tumulosa eDNA absence given x positive qPCR replicates, 1 − ψ(x), and the probability of x positive qPCR replicates given eDNA presence, q (x). Visually estimated presence-absence data were used to augment modeling of C. tumulosa eDNA.

Two chains, thinned at 20 iterations, were run for 2,000 burn-in iterations and 2,000 additional iterations using default priors. Chain convergence was assessed using the Geweke diagnostic (Geweke, 1989) and by visualizing resulting trace plots using the application. Covariate importance was estimated through Bayesian variable selection (Griffin et al., 2020) using an Add-Delete-Swap approach and Pólya-Gamma sampling, automated within the eDNA RShiny application (Diana et al., 2021). Covariates were considered important predictors if their posterior inclusion probability (PIP) values were > 0.5 (i.e., appearing in more than 50% of model iterations) and confidence intervals did not include zero.

Results

A total of 112 membrane samples (not including negative controls) were collected from 34 sites across three atolls in PMNM. Of those samples, 92 were passively collected using PEDS (20 MCE filters vs. 20 cotton rounds with the remaining 52 deployed as cotton on SCUBA divers). For active filtration, 20 MCE filters were actively filtered on vacuum pumps. No amplification occurred in the 30 no-template controls or 29 field and equipment blanks (Fig. S1) and eDNA matrix samples did not exhibit any clear patterns associated with inhibition (linear model: df = 1, F = 2.92, p = 0.10, Table S2). Efficiencies across qPCR plates ranged from 86–96%. When compared to direct visual surveys of presence-absence, assigning naïve occupancy from raw qPCR data correctly inferred C. tumulosa presence at all but two sites (Fig. S2): C. tumulosa eDNA was not detected at PHR05 even though surveyors marked it as present (<1%) but was detected at PHR06, where surveyors marked it as absent. Therefore, naïve occupancy from raw qPCR data was 0.56 (19/34 sites).

Method comparison

Among sites where passive and active methods were directly compared using MCE membranes, actively filtered samples had significantly greater absolute eDNA starting quantities than passively collected ones (Wilcoxon signed-rank test: V = 7368, p < 0.001, Fig. 2A), yet both methods produced the same naïve occupancy estimate (0.8, or 8/10 sites) based on raw qPCR data (Fig. S3, Table S1).

Among sites where passive membrane materials were compared, cotton rounds had only marginally greater eDNA starting quantities (Wilcoxon signed-rank test: V = 114, p = 0.08, Fig. 2B). Both membrane types produced the same naïve occupancy estimate (0.42, or 5/12 sites, Fig. S4, Table S1).

Site-occupancy detection modeling

We assessed the relationships between continuous covariates using Pearson’s correlation which identified a significant correlation between site position X-Y coordinates (t =  − 23.6, df = 32, r =  − 0.97, p < 0.01) as well as between site depth and exposure duration (t = 6, df = 32, r = 0.73, p < 0.01). Following the removal of site longitude coordinates and site depth, the following covariates were retained for further analysis: C. tumulosa benthic cover, exposure duration, and latitude.

The hierarchical modeling framework provided estimates of occupancy (ψ), sample capture (θ11), qPCR replicate detection (p11), and error rates associated with passive eDNA collection methods. Model covariates that significantly influenced assay performance are summarized in Table 1. Among the three passive approaches tested, the use of MCE membranes positively influenced p11 detections; cotton rounds had overall lower rates of p11 (Fig. 3). C. tumulosa relative abundance influenced the probability of eDNA occupancy (ψ) and rate of false-positive test errors (p10). Sites with higher abundance of C. tumulosa were more likely to be occupied by target eDNA but also more likely to produce false-positive replicate detections. In terms of eDNA capture, cotton rounds on SCUBA divers produced the most variable estimates of θ11 (Fig. 3), although this method was not found to be an overall important predictor of eDNA sample capture (Table 1).

Figure 2 Efficiency-corrected quantitative polymerase chain reaction (qPCR) starting quantities (N0) of Chondria tumulosa eDNA.

Samples were collected using (A) active and passive filtration methods on analytical mixed-cellulose ester (MCE) research filters and (B) passive eDNA samplers (PEDS) with cotton rounds or MCE filters.

Table 1 Site-occupancy detection model estimates using the passive eDNA assay.

Posterior mean and 95% credible interval (CI) for model regression coefficients: probability of eDNA site occupancy (ψ), true capture (θ11), false-positive inference capture (θ10), true detection (p11), and false-positive test detection (p10). Posterior inclusion probabilities (PIP, or proportion of iterations included in the model) of covariates linked to each parameter are also included. Model covariates were: Chondria tumulosa relative abundance (“%Cover”, estimated from direct visual surveys), site Y coordinates (“Latitude”), passive eDNA sampler (PEDS) deployment method (cotton/buoy, MCE/buoy, and cotton/SCUBA), PEDS exposure duration (“Exposure”), and the interaction of abundance and exposure duration (“%Cover:Exposure”). Coefficients with PIPs > 0.5 and with CIs that do not overlap zero, are bolded.

	Intercept	%Cover	Latitude	Method MCE/Buoy	Method Cotton/SCUBA	Exposure	Cover: Exposure	
ψ	0.58	0.73	0.23	0.33	0.20	−0.07	0.16	
(CI)	(0.37, 0.77)	(0.05, 1.49)	(−0.48, 0.93)	(−0.52, 1.21)	(−0.67, 1.11)	(−0.77, 0.65)	(−0.73, 1.01)	
PIP	1	0.77	0.47	0.50	0.50	0.43	0.47	
θ 11	0.90	0.40	1.95	0.60	−0.28	0.35	0.04	
(CI)	(0.70, 0.99)	(−1.36, 2.75)	(−0.02, 3.77)	(−1.80, 3.03)	(−2.76, 2.21)	(−1.84, 2.16)	(−2.70, 2.72)	
PIP	1	0.38	0.87	0.48	0.48	0.26	0.46	
θ 10	0.07	0.42	−1.23	0.05	0.80	−0.79	0.56	
(CI)	(0.01, 0.21)	(−2.27, 3.19)	(−2.95, 0.47)	(−2.62, 2.50)	(−1.89, 3.26)	(−2.90, 1.55)	(−2.21, 3.24)	
PIP	1	0.43	0.53	0.46	0.46	0.42	0.43	
p 11	0.76	0.83	−1.44	2.64	1.02	1.08	−0.57	
(CI)	(0.54, 0.91)	(−0.51, 2.39)	(−2.82, 0.18)	(1.39, 3.95)	(−1.18, 3.34)	(−1.38, 3.21)	(−2.86, 2.53)	
PIP	1	0.64	0.91	1	1	0.74	0.72	
p 10	0.04	1.68	−1.36	0.12	1.02	0.74	1.14	
(CI)	(0.01, 0.11)	(0.50, 3.94)	(−2.64, 0.03)	(−2.44, 2.49)	(−1.59, 3.42)	(−0.50, 1.92)	(−1.32, 3.59)	
PIP	1	0.95	0.86	0.66	0.66	0.53	0.59	
Notes.

Presence is the estimated probability that C. tumulosa eDNA is present at a site, given all modeled observations. Occupancy (ψ) is the inherent probability that a site is occupied by C. tumulosa eDNA. True capture (θ11) is the probability of a sample containing C. tumulosa eDNA from an occupied site. True detection (p11) is the probability of qPCR replicate detection from a sample containing C. tumulosa eDNA.

Site occupancy increased with C. tumulosa relative abundance (Fig. 4A) and was consistent across exposure durations (15–225 min, Fig. 4B). Rates of eDNA sample capture (θ11) remained unchanged across abundances of C. tumulosa (Fig. 4C), but varied with exposure (Fig. 4D). The rate of qPCR true-positive detections (p11) increased with C. tumulosa relative abundance, and exposure duration (Figs. 4E–4F).

Figure 3 Site-occupancy detection model estimates for methods comparisons.

Posterior mean and 95% confidence intervals for model parameters from site-occupancy detection modeling of passive eDNA collection methods: MCE filters on a passive buoy (“MCE/Buoy”), cotton membranes on SCUBA divers (“Cotton/SCUBA”), and cotton membranes on a passive buoy (“Cotton/Buoy”). Posterior means are colored by direct visual estimates of C. tumulosa presence (closed triangles) or absence (closed circles). Estimates from all 34 sites using cotton membranes (“Cotton Only”) are in black and our previous work using only actively filtered C. tumulosa eDNA across the Hawaiian Archipelago (“Active Only”) (Nichols et al., 2025) is in grey. Occupancy (ψ) is the inherent probability that a site is occupied by C. tumulosa eDNA. True capture (θ11) is the probability of a sample containing C. tumulosa eDNA from an occupied site. True detection (p11) is the probability of qPCR replicate detection from a sample containing C. tumulosa eDNA.

Figure 4 Site-occupancy detection model estimates of passive assay parameters.

Posterior mean and 95% credible intervals for model parameters: probability of Chondria tumulosa occupancy (ψ, inherent probability of eDNA presence based on site characteristics, A–B), sample capture given C. tumulosa eDNA presence at a site (θ11, C–D), and probability of quantitative polymerase chain reaction (qPCR) replicate detection given C. tumulosa eDNA presence in a sample (p11, E–F). Posterior means are colored by direct visual estimates of C. tumulosa presence (orange) or absence (green). Trendlines represent generalized additive model smoothers across benthic cover (range: < 1%–85%) and exposure duration (range: 15–225 min).

Baseline assay performance

Across all surveyed sites, the baseline posterior mean occupancy (ψ) was estimated to be 0.58 (CI [0.37–0.77]), sample capture (θ11) was 0.90 (CI [0.70–0.99]), and qPCR detection (p11) was 0.76 (CI [0.54–0.91], Table 1). The estimated false-positive inference rate during the field sampling stage (θ10) was 0.07 (CI [0.01–0.21]) and the false-positive test rate was 0.04 (CI [0.01–0.11]) during the laboratory analysis stage (p10). False-negative inferences were estimated to be 0.10 (CI [0.01–0.30]) at the sampling stage (1 − θ11) and false-negative tests were 0.24 (CI [0.09–0.46]) at the laboratory stage (1−p11). To gauge the performance of qPCR replication, we modeled the conditional posterior probability of species absence given x qPCR positive amplifications (i.e., the probability of generating a false-positive inference based on x positive qPCR replicates, 1-ψ(x)). The conditional probability of a false-positive inference given three or more positive qPCR replicates was <8%, whereas for all six it was 5% (Fig. 5), as estimated from the occupancy model posterior distributions. Given the presence of C. tumulosa eDNA, the assay is highly likely (85%) to produce three or more positive qPCR replicates.

Figure 5 Assessment of quantitative PCR (qPCR) technical replication.

The conditional probability of Chondria tumulosa eDNA absence at a site, 1-ψ (x), given x positive qPCR replicates and the probability of x positive qPCR replicates given the presence of C. tumulosa eDNA, q (x) from site-occupancy detection modeling of passive eDNA filters.

Discussion

The rapid spread of NIS necessitates highly sensitive and practical survey approaches to facilitate widespread use (Kelly et al., 2014; Dickie et al., 2018). Passive collection of eDNA has emerged as a technique to increase the sampling capacity over conventional filtration methods using vacuum pumps. The numerous fundamental differences between PEDS sampling and the existing eDNA assay makes it difficult to directly compare detection results. Nonetheless, the active filtration of 2-L surface seawater samples provides a benchmark by which to compare the performance of PEDS: site-occupancy detection modeling of eDNA collected using PEDS estimated similar capture rates (θ11), however there was a reduction in qPCR detection rates (p11) and an increase in the probability of false positives at both the field collection (θ10) and laboratory processing (p10) stages (Table S3). This suggests that while passive eDNA capture can be comparable to active filtration, a higher number of qPCR replicates is necessary to differentiate true positive detections from false positive inferences and tests. Importantly, increasing the number of qPCR replicates is a more straightforward and scalable approach than implementing active filtering methodologies, which requires more equipment, coordination, and a greater sampling effort (Furlan et al., 2016).

Our findings demonstrate that passive eDNA sampling provides an efficient, cost-effective, and flexible method for detecting NIS. We found that briefly submerging cotton rounds in seawater was sufficient to detect C. tumulosa along its colonization front in abundances far too low (≤ 1%) to be detected by an untrained visual observer. In PMNM, passive eDNA sampling had equal clinical sensitivity compared to active filtration, detecting target eDNA at all of the same sites, and can be paired with species-specific assays for rapid NIS biomonitoring. Despite being just as sensitive to target eDNA at the site level, PEDS had lower performance, exhibiting greater error rates and recovering fewer DNA copies (which we discuss in detail below). The equipment required for passive eDNA collection is more cost-effective and less prone to contamination, as it involves fewer processing steps (Goldberg et al., 2016; Minamoto et al., 2016; Djurhuus et al., 2017). Furthermore, PEDS can be deployed in various ways, allowing flexibility depending on the specific needs of the end-user. For example, in deeper sites PEDS were attached to SCUBA divers, enabling them to passively survey for C. tumulosa while multitasking with limited bottom time. Furthermore, our passive sampling approach allowed assay-guided sampling to maximize the efficiency and capacity of limited field days in remote locations. Coupled with advancements in field-ready DNA extraction methods (e.g., Guevara et al., 2018; Stanton et al., 2019), portable qPCR machines (e.g., Doi et al., 2021), and diagnostic isothermal amplifications (e.g., Blin et al., 2023; Jothinarayanan et al., 2024), passive eDNA sampling has the potential to generate detection results in near-real-time, eliminating the need for dedicated laboratory space and the hassle of operating and cleaning filtration equipment.

Passive eDNA collection and molecular detection of C. tumulosa was not perfect. Although passive methods reduce the number of filtration steps that could introduce contaminants, we observed an increase in the probability of false positives compared to conventional filtration, despite no apparent contamination in negative controls. The term “false positive” can refer to both site-level errors (e.g., eDNA detection where the species is presumed absent) and sample-level errors (e.g., contamination or PCR artifacts). Following Darling, Jerde & Sepulveda (2021), we distinguish among error types within the eDNA sampling hierarchy: “presumed-positive” and “presumed-negative” refer to eDNA-based inferences lacking independent validation; a “false-positive inference” occurs when a presumed-positive site is contradicted by direct observations; a “false-positive test” results from laboratory errors, and a “false-negative test” arises when eDNA fails to detect a species known to be present.

We explicitly modeled error probabilities at each level to address uncertainty in molecular detections. Significantly lower DNA concentrations recovered from PEDS relative to active filtration may increase the relative risk of contamination in low-template samples (Goldberg et al., 2016). One false-positive inference (PHR06, Fig. S2), which produced nine out of 12 positive qPCR replicates, was estimated to have a 28% probability of C. tumulosa eDNA presence even though surveyors did not observe the alga (Table S1). False-positive inferences are not necessarily due to sample contamination (Darling, Jerde & Sepulveda, 2021). The cryptic growth of C. tumulosa could lead to visual misidentifications which can explain both false-positive inferences (e.g., by overlooking the target taxa) and false-negative detections (e.g., by misidentifying the target taxa for species such as Palisada parvipapillata or Laurencia spp.). Additionally, false-positive tests could result from contamination via exposure of membranes to air (e.g., Clare et al., 2021) or trace amounts of eDNA present in the field (e.g., from contaminated dive gear).

To minimize false-positive tests, membranes are handled with bleach-decontaminated forceps and sterile gloves, and PEDS should remain sealed in plastic bags until immediate deployment. While contamination from non-target sources in cotton membranes is less concerning for species-specific qPCR assays, UV-sterilization of membranes before deployment can further mitigate contamination risks. Nonetheless, most of these issues can be mitigated by increasing the number of qPCR replicates, which reduced modeled false-positive inferences. When using six positive qPCR replicates as the detection threshold, the likelihood of a false-positive inference was reduced from 11% to 5% (Fig. 5). This confirms that increasing qPCR replication can reduce erroneous inferences (Ficetola et al., 2015; Yamamoto et al., 2017; Stevens et al., 2024), allowing for 95% certainty in the detection of C. tumulosa. The costs of incorporating additional qPCR replicates are minimal compared to the substantial effort required for conducting the same study using active filtration methods.

Another consideration is minimizing errors due to lower eDNA yields from PEDS. Passive samples consistently yielded lower concentrations of C. tumulosa eDNA compared to actively filtered samples. This result aligns with findings from similar studies in aquatic environments (Kirtane, Atkinson & Sassoubre, 2020; Bessey et al., 2021; Van der Heyde et al., 2023). Thus, passive methods may be prone to higher rates of false-negative detections. In fact, false-negative detection rates (1-p11) were four times greater than those in our previous study using only active filtration methods (24% vs. 6%) (Nichols et al., 2025, Table S3). To improve true detection rates, efforts should be made to increase DNA yield at either the field or laboratory stages. For instance, allowing passive membranes to be exposed to the environment for longer periods could enhance the capture of target eDNA (Chen et al., 2022). However, our modeling results suggest that extending exposure time does not necessarily improve C. tumulosa eDNA capture rates (θ11; Fig. 4D), as similarly reported in other studies investigating PEDS sampling duration (e.g., Bessey et al., 2022). Although, longer deployments may improve detection within replicates (p11; Fig. 4F), potentially due to increased eDNA accumulation on the membrane. Although cotton rounds can produce greater relative DNA yields than MCE filters (Fig. 2B), this difference was not statistically significant. The use of MCE filters in PEDS increased qPCR detection rates (p11), but often resulting in detections that barely exceeded the threshold of the assay (Fig. S4). Even though natural fibers tended to have lower DNA yields in PEDS trials (Bessey et al., 2022), we recommend cotton rounds to minimize cost and maximize the surface area for eDNA adsorption.

Directly comparing the performance of two fundamentally different substrates is challenging due to differences in processing and sample volume. Specifically, only half of each MCE filter was processed (and the other half was archived), representing 50% of the total filter area and a 1-L effective sample volume per reaction (Nichols et al., 2025). In contrast, cotton rounds were further subsampled to fit into two mL tubes for DNA extraction (representing ∼8.25% of the total membrane area). This discrepancy complicates absolute quantification, as the effective sample volume from PEDS remains uncertain. Furthermore, the pairing of PEDS with actively pumping water could increase the volume of water flowing over nearby passive samplers. These potential effects were minimized by separating the pump inflow hose from PEDS by 0.5 m and operating the peristaltic pump at its lowest setting (0.13 L min−1). Pooling multiple fractions of cotton rounds and further optimizing the eDNA extraction protocol could potentially increase qPCR detection rates (Hunter et al., 2019). In shallow coral reef environments, PEDS captured C. tumulosa eDNA as effectively as active filtration methods, albeit with lower target DNA yield per reaction, which can be somewhat offset by using absorbent cotton rounds. Given their low cost and widespread availability, cotton rounds are sufficiently sensitive and a highly practical option for scalable passive eDNA sampling. Further research into the binding affinities of eDNA and associated particulate matter (Turner et al., 2014) on membranes such as cotton is needed to better understand how passive collection can optimize the detection of low-abundance taxa like C. tumulosa.

Conclusions

Passive eDNA sampling is gaining traction as a tool to survey biodiversity using various substrates, including air, plants, biofilms, and aquatic filter feeders (Valentin et al., 2018; Mariani et al., 2019; Clare et al., 2021; Rivera et al., 2022). However, PEDS have had varying success in detecting rare aquatic taxa (Bessey et al., 2022; Jeunen et al., 2022). Actively filtering water remains the preferred method for detecting C. tumulosa across sites with varying depths and abundances (Nichols et al., 2025). Nevertheless, our study suggests that with relatively brief 15-minute deployments, PEDS can reliably detect C. tumulosa even at low abundances, provided the membranes remain close to the benthic habitat. While marginally less effective than traditional methods, the rapid eDNA collection time and cost savings of passive sampling can justify its use (Yan et al., 2024), especially when increasing sampling capacity and improving biomonitoring inferences are a top priority (Schmidt et al., 2013; Willoughby et al., 2016; Zinger et al., 2019; Buxton et al., 2021). Advancing passive eDNA collection techniques has the potential to make biomonitoring of aquatic NIS more accessible, cost-effective, and widespread.

Supplemental Information

Supplemental Information 1 Field sites from shallow coral reef atolls in Papahānaumokuākea Marine National Monument, Hawaiʻi, USA

Passive eDNA samplers (PEDS) were deployed at three atolls: Hōlanikū, or Kure Atoll, Kuaihelani, or Midway Island, and Manawai, or Pearl & Hermes Atoll. Chondria tumulosa benthic cover was estimated within an approximately 314 m2 area by two surveyors at each site. Exposure duration was 15 min. for all deployments of PEDs on stationary buoys, but varied when attached directly to SCUBA divers. The number of qPCR technical replicates with positive quantification cycle (Cq) values are listed for both biological samples per site. Posterior probabilities and credible intervals (CI) are listed for parameter estimates from site-occupancy modeling. The number of positive qPCR detection replicates per sample is listed. The method of PEDS deployment is also provided (active MCE vs. passive MCE, passive cotton vs. passive MCE on buoys, or passive cotton on SCUBA).

Supplemental Information 2 Linear model for testing of qPCR inhibition

Results from an analysis of covariance (ANCOVA) testing for PCR inhibition by comparing Cq values and log-transformed DNA starting quantities (log10(StartingQuantity)) between synthetic DNA standards and standards spiked with eDNA matrix (“Template Type”).

Supplemental Information 3 Comparison of model parameter estimates from the previous actively-filtered assay (Nichols et al., 2025) and the membranes in passive environmental DNA samplers (PEDS)

Posterior mean and 95% credible interval (CI) for model regression coefficients: probability of true capture (θ11), false-positive inference capture (θ10), true detection (p11), and false-positive test detection (p10). The probabilities of false-negatives at the field stage (1-θ11) and laboratory stage (1-p11) are the complements of θ11 and p11, respectively.

Supplemental Information 4 Amplification from positive and negative control samples

Quantitative polymerase chain reaction (qPCR) amplification curves from control samples: positive serial dilution standards from a stock solution of Chondria tumulosa tissue (“Standard”), qPCR no-template controls (“NTC”, n = 30), and field/equipment blanks (“EB”, n = 29). The mean fluorescence quantification threshold (±SE) is marked with a dashed grey line.

Supplemental Information 5 Environmental DNA (eDNA) quantitative polymerase chain reaction (qPCR) amplification curves from passive eDNA sampler (PEDS) deployments

Sites were from Hōlanikū (“H”, or Kure Atoll), Kuaihelani (“K”, or Midway Island), and Manawai (“PHR”, or Pearl & Hermes Atoll). Amplification (relative fluorescence units, RFU) of Chondria tumulosa eDNA is marked with circles (representing each individual qPCR replicate) and a solid line generalized additive model smoother of triplicate PCR reactions among water samples from each site. Points and lines are colored blue if they exceed the average (±SE) threshold of detection. Control samples consisting of: positive C. tumulosa tissue extraction serial dilutions (“Standard”), qPCR no-template controls (“NTC”), and equipment blanks (“EB”). Sites with confirmed detections from direct visual surveys are marked with a ().

Supplemental Information 6 Raw quantitative PCR (qPCR) amplification curves between active and passive methods

Comparison of environmental DNA (eDNA) quantitative polymerase chain reaction (qPCR) amplification curves from passive eDNA sampler (PEDS) deployments (“Passive MCE”, solid black line) and active filtration of 2-L water samples (“Active MCE”, solid grey line). Sites were from Hōlanikū (“H”, or Kure Atoll) or Kuaihelani (“K”, or Midway Island). Amplification (relative fluorescence units, RFU) of Chondria tumulosa eDNA is marked with a best-fit generalized additive model smoother of triplicate PCR reactions among water samples from each site. The mean fluorescence quantification threshold (±SE) is marked with a dashed grey line.

Supplemental Information 7 Raw quantitative PCR (qPCR) amplification curves between tested membrane materials

Comparison of environmental DNA (eDNA) quantitative polymerase chain reaction (qPCR) amplification curves from passive eDNA sampler (PEDS) deployments using cotton membranes (“Passive Cotton”, solid black line) or mixed cellulose ester analytical filters (“Passive MCE”, solid grey line). Sites were from Hōlanikū (“H”, or Kure Atoll) or Kuaihelani (“K”, or Midway Island). Amplification (relative fluorescence units, RFU) of Chondria tumulosa eDNA is marked with a best-fit generalized additive model smoother of triplicate PCR reactions among water samples from each site.

We would like to thank Elaine Johnson, Jonathan Plissner, Tammy Summers, and Morgan Walter for their support with field sampling and logistics at the Midway Atoll National Wildlife Refuge. Kaua‘oa Fraiola, Brian Hauk, Chelsie Counsell, Sydney Luitgaarden, Kimberly Fuller, Colt Davis, and the crew of the Imua and Kahana II supported in-field sampling logistics elsewhere in the Monument. Alison Sherwood and Monica Paiano supplied C. tumulosa DNA tissue template extractions. Randall Kosaki and Jason Leonard donated ship time, marine assets, and the staff time required for data collection. Portions of this manuscript were revised for clarity and readability with AI-assisted language tools (ChatGPT, OpenAI). The scientific content and interpretations were developed and drafted solely by the authors.

Additional Information and Declarations

Competing Interests

Author Contributions

Field Study Permissions

Data Availability

The authors declare there are no competing interests.

Patrick K. Nichols conceived and designed the experiments, performed the experiments, analyzed the data, prepared figures and/or tables, authored or reviewed drafts of the article, and approved the final draft.

Peter B. Marko conceived and designed the experiments, authored or reviewed drafts of the article, and approved the final draft.

The following information was supplied relating to field study approvals (i.e., approving body and any reference numbers):

Field work was conducted under Papahānaumokuākea Marine National Monument Co-Trustees permits PMNM-2023-01 and PMNM-2024-001.

The following information was supplied regarding data availability:

qPCR data are available at Zenodo:

Patrick Nichols. (2025). pknichols/passive_eDNA: Data files added (v1.0.1). Zenodo. https://doi.org/10.5281/zenodo.14991498.

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
