# Peer review of "No pump, no problem: evaluating passive eDNA sampling for marine biomonitoring of a nuisance macroalga"

_PeerJ, doi:10.7717/peerj.19939_

## Round 0.1 · original submission · Minor Revisions

Dear authors, two reviewers have read and commented on your manuscripts. They agree that the article is well written and has a solid scientific background. However, they provided some small comments to improve it.

·

Basic reporting

Overall the manuscript is well-written and clear. Some improvements in consistency regarding the general conclusion on the effectiveness of PEDS. For example, the abstract contains contradictory statements – "methods performed equally [well]," which is contradicted by the statements on L27-28. Later in the manuscript, there are similar contradictions, for example on L381 the PEDS approach is described as “nearly as sensitive” but L420-421 describe higher (by 2X) false-negative rates. Please ensure the descriptive language is consistent with the results and conclusions.

Experimental design

Model may have to be re-run to reflect a different definition of "false positive;" see details in Part 3.

Validity of the findings

I would dispute the definition of “false positive” here, e.g. as presented on L337-338, L397-398, and L411. As mentioned on L407, C. tumulosa DNA could be present at a site in which it was not visually observed, which means the assay worked as intended (and is highly sensitive). For NIS work, the presence of target eDNA is likely still of high interest because it could signal the impending arrival of the species and provide opportunities to prepare for / mitigate the invasion. So, I would suggest changing the term “false positive” to something else that appropriately the describes the absence of a corresponding visual detection, like “eDNA-only” or similar. This change may also require re-running the model; if I understand it correctly, the model relies on “opportunistic visual surveys” to infer false positives and false negatives (L262-263), which is not necessarily appropriate for the reasons described above. I defer to the authors’ modeling expertise to make that decision.

Additional comments

L22: equally “well”
L164-166: A minor thought/consideration based on the description of this setup - could the PEDS have benefited from the water flow facilitated by the active pumping?
L185-197: I didn't see any description of decontaminating the cotton rounds, which might have lots of terrestrial DNA. In this case it wouldn't necessarily be a contamination issue for a marine macroalga, but for metabarcoding or detection of insects or fungi this could be an issue? I would suggest UV-sterilization if possible before field deployments.
L315: This should be absolute amounts, not relative amounts? If describing total eDNA not just target gene copies.
L424: Cite Figure 4 here.
Figure 5: Probability of absence is a little counter-intuitive. If it were written out on the y-axis that would help.

·

Basic reporting

Dear Editor,
I carefully reviewed the manuscript "No pump, no problem: evaluating passive eDNA sampling for marine biomonitoring of a nuisance macroalga" by Patrick Nichols and colleagues.The manuscript reports a comparison between passive environmental DNA (eDNA) collection methods and conventional active filtration for the dectection of the cryptogenic macroalga Chondria tumulosa in in Hawaii.

The manuscript is very well written, clear, professional, and easy to follow. In my opinion the authors provide sufficient ecological background and methodological context, and the literature cited is relevant, current, and well-integrated into the narrative. The structure of the article is also very good and clear. and the figures and tables of high quality. The only minor suggestion is that it maybe would be helpful to include the legends directly within the supplementary figure files (e.g., embedded in the PNGs) and to provide the legends at the top of each supplementary table for clarity

Research permits numbers are reported in the text, but I was not able to check if they are appropriate because it seems to me that they are not publicly available.

Overall, in my opinion, the manuscript is very interesting and represents an appropriate "unit of pubblication".

Experimental design

The manuscript fits the Aims and Scope of the Journal, with research questions well defined and relevant.

The investigation does not pose ethical concerns and has been conducted to a high standard. I have only one concern regarding the PCR inhibition testing, which was carried out "by comparing the standard curve efficiencies of sample DNA matrix against C. tumulosa synthetic DNA (L238–242)". I believe this method is suboptimal compared to the use of an internal positive control (IPC)—see, for instance, Goldberg et al. 2016 (https://doi.org/10.1111/2041-210X.12595)—and that insufficient detail is provided in both the Materials and Methods and the Results sections. Why was an IPC not used? What is meant by “Cq values not consistent with the synthetic standard curve” at L242? On how many samples was inhibition tested? Additionally, the statement that “eDNA samples did not exhibit any clear patterns associated with inhibition” (L305–306) is unclear, and some further explanation/reporting would be helpful.

Validity of the findings

In my opinion findings are valid and very interesting, underlying data have been provided and conclusions are ok.

Additional comments

1. Please check the number of sites in L302: why 32 and not 34?
2. L304: maybe change to “further 20 MCE filters”?
3. L317 and L321: perhaps cite Table S1 here as well, where the number of sites used for comparing passive/active methods and membrane materials is clearly indicated.
4. L397–399: Was the amount of DNA lower in PEDS than with filtration? If so, this could increase the risk of contamination. I suggest adding a brief comment on this point—the result may not be so “unexpected”.

---

## Round 0.2 · accepted · Accept

Dear Authors

I am happy to inform you that your paper has been accepted. The reviewers agree that you answer all their questions.

·

Basic reporting

no comment

Experimental design

no comment

Validity of the findings

no comment

Additional comments

The authors have met all my minor concerns and improved an already excellent manuscript. I look forward to seeing it in publication.

·

Basic reporting

The authors did a good job in responding to my issues and I do not have further questions.

Experimental design

The authors did a good job in responding to my issues and I do not have further questions.

Validity of the findings

The authors did a good job in responding to my issues and I do not have further questions.

Additional comments

None